# Type II Heterojunction Formed between {010} or {012} Facets Dominated Bismuth Vanadium Oxide and Carbon Nitride to Enhance the Photocatalytic Degradation of Tetracycline

**DOI:** 10.3390/ijerph192214770

**Published:** 2022-11-10

**Authors:** Xiaojing Zhang, Xianglun Xie, Jianan Li, Dongfang Han, Yingming Ma, Yingying Fan, Dongxue Han, Li Niu

**Affiliations:** 1Center for Advanced Analytical Science, Guangzhou Key Laboratory of Sensing Materials & Devices, School of Chemistry and Chemical Engineering, Analytical and Testing Center, Guangzhou University, Guangzhou 510006, China; 2Guangdong Provincial Key Laboratory of Psychoactive Substances Monitoring and Safety, Anti-Drug Technology Center of Guangdong Province, Guangzhou 510230, China

**Keywords:** bismuth vanadium oxide, carbon nitride, photocatalytic, tetracycline, facets, heterojunction

## Abstract

Both type II and Z schemes can explain the charge transfer behavior of the heterojunction structure well, but the type of heterojunction structure formed between bismuth vanadium oxide and carbon nitride still has not been clarified. Herein, we rationally prepared bismuth vanadium oxide with {010} and {012} facets predominantly and carbon nitride as a decoration to construct a core-shell structure with bismuth vanadium oxide wrapped in carbon nitride to ensure the same photocatalytic reaction interface. Through energy band establishment and radical species investigation, both {010} and {012} facets dominated bismuth vanadium oxide/carbon nitride composites exhibit the type II heterojunction structures rather than the Z-scheme heterojunctions. Furthermore, to investigate the effect of type II heterojunction, the photocatalytic tetracycline degradations were performed, finding that {010} facets dominated bismuth vanadium oxide/carbon nitride composite demonstrated the higher degradation efficiency than that of {012} facets, due to the higher conduction band energy. Additionally, through the free radical trapping experiments and intermediate detection of degradation products, the superoxide radical was proven to be the main active radical to decompose the tetracycline molecules. Therein, the tetracycline molecules were degraded to water and carbon dioxide by dihydroxylation-demethylation-ring opening reactions. This work investigates the effect of crystal planes on heterojunction types through two different exposed crystal planes of bismuth vanadate oxide, which can provide some basic research and theoretical support for the progressive and controlled synthesis of photocatalysts with heterojunction structures.

## 1. Introduction

Tetracycline (TC) is known as the second most widely used organic compound of broad-spectrum antibiotics in the world and is also applied extensively in medical and farming applications [1,2]. TC and their derivatives are stable in the environment, which tends to damage the ecosystem and affect human health, even increasing drug resistance in humans as well as in plants and animals [3]. In recent years, the abuse of TC has led to frequent detection in wastewater emissions, surfaces and groundwater, and even in soil environments [4,5,6,7,8]. Therefore, it is urgent to find an efficient and green way to solve the problem of TC pollution. Photocatalytic semiconductor technology, with the utilization of solar energy, has high catalytic oxidation activity and is environmentally friendly, which has also proven to be a very viable strategy and has attracted the attention of researchers worldwide [9,10,11,12,13].

Many semiconductor materials such as titanium dioxide [14], zinc oxide [15], cuprous oxide [16], bismuth vanadium oxide (BiVO_4_) [17], and carbon nitride (g-C_3_N_4_) [18,19] have been successfully used in the photocatalytic degradation of organic pollutants. Among these materials, it has been proven that BiVO_4_-based platforms with different exposed crystalline planes were able to drive electrons and holes to distinct dominant crystal faces due to their variant surface energy levels [20]. Such advantages will greatly facilitate the separation of electrons and holes, thus availably improving the performance of photocatalytic semiconductors. Nevertheless, limited by the local distance of transmission, photo-generated electrons and holes excited by light irradiation on individual semiconductors still tend to combine with each other [21]. In this case, through constructing heterojunction constructions between BiVO_4_ and other semiconductors, the photogenerated carriers can be effectively separated by band matching [22,23,24].

However, for the configurations of BiVO_4_ heterojunction complex photocatalysts, there are many controversies. For example, Wang et al. [25] prepared BiVO_4_/g-C_3_N_4_ heterojunction composites, following the type II model of electron transfer, demonstrating increased photocatalytic degradation performance on TC. However, Li et al. [26,27,28] suggested that the electron transfer in BiVO_4_/g-C_3_N_4_ heterojunctions followed a Z- scheme model. The reason why the controversy is emerging is that, whether following the type II or Z-scheme heterojunction structure, both the carrier separation behavior and photocatalytic performance improvement can be well explained. However, for the sake of scientific rigor, the accurate and actual heterojunction-type structure should be figured out. Since the exposed crystal facets of semiconductors can affect their energy band constructures [29,30,31], which may further affect the heterojunction establishments, thus the different facets of exposed BiVO_4_/g-C_3_N_4_ composites should be investigated.

In this paper, the separately dominant {010} and {012} facets exposed BiVO_4_ crystals are synthesized and combined with g-C_3_N_4_ to form heterojunction structures ({010}BiVO_4_/g-C_3_N_4_ and {012}BiVO_4_/g-C_3_N_4_). Due to the flat and ordered crystal facets, the g-C_3_N_4_ sheets could be tightly packed around crystals of {010}BiVO_4_ and {012}BiVO_4_, impeding the direct contact of reactants by {010}BiVO_4_ and {012}BiVO_4_. This simplifies the carrier flow investigation and influence analysis of the TC degradation. Thus, through the establishment of energy band, the changes in the type of photogenerated radicals and the degradation performances of TC, both {010}BiVO_4_/g-C_3_N_4_ and {012}BiVO_4_/g-C_3_N_4_ composites proven to be the type II heterojunction structures. Additionally, the {010}BiVO_4_ with a more negative conduction band than {012}BiVO_4_ can facilitate a higher fabrication amount of superoxide radical (O_2_^•−^). In the BiVO_4_/g-C_3_N_4_ system, the O_2_^•−^ radical exhibits a much higher efficiency than the hydroxyl radical (^•^OH) for TC degradation. This work is excepted to settle the contradictory phenomena on the heterojunction structures of BiVO_4_/g-C_3_N_4_ composites.

## 2. Experimental

### 2.1. Materials

All the chemical materials and reagents are not purified in any further way. Bismuth nitrate pentahydrate (Bi(NO_3_)_3_·5H_2_O, 98%), ammo-nium vanadate (NH_4_VO_3_, 99.0%), bismuth oxide (Bi_2_O_3_), and vanadium oxide (V_2_O_5_) were purchased from Innochem. Moreover, tetracycline (C_22_H_24_N_2_O_8_), sodium dodecylbenzene sulfonate (SDBS), urea (CH_4_N_2_O, 99.0%), isopropyl alcohol (C_3_H_8_O,99.5%), Sodium sulfate anhydrous (Na_2_SO_4_, 99.0%), *p*-Benzoquinone (C_6_H_4_O_2_), and Triethanolamine (C_6_H_15_NO_3_) were purchased from Macklin. Deionized water with a resistivity of 18.25 MΩ cm^−1^ was used in all experiments.

### 2.2. Synthesis of the g-C_3_N_4_

At a rate of 0.5 °C/min, 10 g urea in alumina crucible (with cover) was heated to 550 °C in air atmosphere and kept at this temperature for 3 h. Afterward, it was cooled naturally to room temperature, and the yellowish powder was then collected.

### 2.3. Preparations of {012}BiVO_4_ and {012}BiVO_4_/g-C_3_N_4_

For the synthesis of {012}BiVO_4_, 2 mmol bismuth nitrate, 200 mg SDBS, 1.25 M nitric acid solution, and 2 mmol ammonium vanadate were added to 100 mL of deionized water and stirred for 30 min. The solution was then transferred to a polytetrafluoroethylene hydrothermal reactor and placed in an oven at 150 °C for 12 h. After completion of the reaction, the reactor was naturally cooled to room temperature, and the product was collected by filtration and washed several times with deionized water.

For the preparation of {012}BiVO_4_/g-C_3_N_4_, the prepared g-C_3_N_4_ was added to the precursor solution after the stirring process, and the other steps were the same as above.

### 2.4. Preparation of {010}BiVO_4_ and {010}BiVO_4_/g-C_3_N_4_

For the synthesis of {010}BiVO_4_, 2.5 mmol of bismuth oxide and vanadium oxide were added to 25 mL of nitric acid solution (0.5 M) and subsequently stirred on a magnetic stirrer for 96 h. Then the product was collected by filtration and washed several times with deionized water.

For the preparation of {010}BiVO_4_/g-C_3_N_4_, the prepared {010}BiVO_4_ was dispersed into an ethanol solution, followed by the addition of g-C_3_N_4_/ethanol solution (20 mg/mL) and stirred at room temperature for 24 h. The subsequent steps were then carried out for the preparation of {010}BiVO_4_.

### 2.5. Structure and Characterization of Materials

The crystal phase and purity of the catalyst were determined by X-ray diffractive apparatus (XRD, Palytical X’Pert Powder) of Panaco. The microstructure and elements of the powder samples were analyzed by field emission scanning electron microscopy (SEM, JSM-7001F) and energy dispersive X-ray spectroscopy (EDX, JSM-701F). The structural characteristics of the samples were observed through a transmission electron microscope (TEM, JSM-2100F) made by Hitachi. The surface element composition and existence state of the synthesized materials were analyzed by X-ray photoelectron spectroscopy (XPS, Thermo Fisher Scientific K-Alpha). The chemical bonds and groups in the samples were analyzed and determined by Fourier infrared spectroscopy (FT-IR, Thermo 6700) from Thermo Field Company. Chemical bonds and groups in the samples were analyzed. The hydroxyl radical and superoxide radical of the samples were analyzed by electron paramagnetic resonance (EPR, A300) spectrometer of Brook Corporation. The comparative area of the samples was measured by the BET tester (BET, ASAP2460) of Mack Company. The intermediate products of the photocatalytic degradation of TC were tested by waters single quadrupole liquid mass spectrometry (LC-MS, ZQ2000). The optical absorption properties of the synthesized materials were characterized by UV-Vis diffuse reflectance spectroscopy (UV-VIS DRS, U-3900) from Hitachi, which scanning range was 200–800 nm, and BaSO4 powder was used as a reference.

### 2.6. Photocatalytic Activity Experiment

100 mL of TC aqueous solution with a concentration of 50 mg L^−1^ was measured and introduced into a 250 mL reactor, and 20 mg of the sample was then added. After evenly dispersed, the sample was stirred in the dark for 30 min. During this period, 3.5 mL of the above solution was taken out every 15 min. After being filtered by a microporous membrane, the clear solution was utilized as analyte, and the adsorption rate of TC was measured using Agilent’s UV/Vis spectrophotometer (Carry 60). After the adsorption and desorption phases were balanced, the above solution was placed under a 300W xenon lamp and illuminated for 60 min. During this period, 3.5 mL of the above solution was taken out every 15 min and filtered, and the concentration of TC was measured by ultraviolet/visible spectrophotometer, and the light absorption intensity at 357 nm was recorded. The degradation rate of TC was calculated by Formula (1).
(1)$$\mathrm{Degradation}\;\mathrm{rate}\;(\%)=(\frac{C_{0}-C}{C_{0}})\;\times \;100\%$$
where $C_{0}$ is the absorbance of the TC when adsorption equilibrium is reached, and C is the absorbance of the TC measured by timed sampling.

### 2.7. Photoelectrochemical Tests

The electrochemical workstation (CHI920D) was used for photoelectric test. Ag/AgCl was employed as reference electrode, platinum electrode as counter electrode, and sodium sulfate (Na_2_SO_4_) with 1 M electrolyte as electrolyte. Preparation of working electrode: 1 mg sample was dispersed in 1 mL deionized water, and ultrasound was carried out for 20 min. 100 μL of the dispersion was then measured with a pipette gun and dropped onto the cleaned FTO conductive glass. The photochemical and electrochemical properties of the samples were tested after natural air drying.

### 2.8. Photocatalytic Trapping Agent Experiment

Isopropyl alcohol (IPA; 0.1M), triethanolamine (TEOA; 0.1 M), and benzoquinone (BQ; 0.2 mM) were added into TC solution (100 mL; 50 mg/L) mixed with 20 mg catalyst, after adsorption equilibrium, the illumination was then performed. During this period, 3.5 mL of the above solution was taken out every 15 min, and its absorbance at 357 nm was recorded with an ultraviolet/visible spectrophotometer. Finally, the degradation rate was calculated according to Formula (1).

## 3. Results and Discussion

### 3.1. Characteristics

Photocatalysts of {010}BiVO_4_ and {012}BiVO_4_ were separately prepared through the wet chemistry method and hydrothermal method, according to the previous reports [17,32]. g-C_3_N_4_ was prepared by a direct calculation method as described in the experimental section. The different loading ratios between {010}BiVO_4_ and g-C_3_N_4_ were via the physical hybrid approaches by long-time stirring operations; meanwhile, the {012}BiVO_4_/g-C_3_N_4_ composites were fabricated by adding different amounts of the prepared g-C_3_N_4_ into the precursor solutions of {012}BiVO_4_. After the synthesis, the architectures and morphologies of {010}BiVO_4_, {012}BiVO_4_, g-C_3_N_4_, {010}BiVO_4_/g-C_3_N_4_, and {012}BiVO_4_/g-C_3_N_4_ were investigated by SEM and TEM. The pure {010}BiVO_4_ sample displays a sheet shape with a length of ca. 1 μm (Figure 1a), while the {012}BiVO_4_ depicts a near-octahedral shape with a size of about 5 μm (Figure 1b). The morphology of g-C_3_N_4_ exhibits a sheet structure, as shown in Figure 1c,d. To verify the successful loading of g-C_3_N_4_ onto BiVO_4_, further SEM tests of {010}BiVO_4_/g-C_3_N_4_ and {012}BiVO_4_/g-C_3_N_4_ were performed. Since the {010}BiVO_4_/0.2g-C_3_N_4_ and {012}BiVO_4_/0.1g-C_3_N_4_ equipped the highest TC degradation performances (discussed in the section of photocatalytic performance) compared to other {010}BiVO_4_/xg-C_3_N_4_ and {012}BiVO_4_/xg-C_3_N_4_ composites, the {010}BiVO_4_/0.2g-C_3_N_4_ and {012}BiVO_4_/0.1g-C_3_N_4_ were taken as instances to investigate the morphologies and structures. Therein, the x values from {010}BiVO_4_/xg-C_3_N_4_ and {012}BiVO_4_/xg-C_3_N_4_ represent the molar ratio of g-C_3_N_4_ to BiVO_4_. As shown in Figure 1e–h, the two hybrids of {010}BiVO_4_/0.2g-C_3_N_4_ and {012}BiVO_4_/0.1g-C_3_N_4_ retained their flake or octahedral shapes unchanged and were almost in the same size. Magnifying the image of {010}BiVO_4_/0.2g-C_3_N_4_ based on TEM tests (Figure 1i), a nanosheet shell of g-C_3_N_4_ is observed to be evenly covered on the surface of {010}BiVO_4_. For {012}BiVO_4_/g-C_3_N_4_, g-C_3_N_4_ is also found to be homogeneously distributed on the surface of {012}BiVO_4_ with a tiny nanosheet structure (Figure 1h), indicating that {012}BiVO_4_ is completely wrapped in a g-C_3_N_4_ sheet. Further chemical element distribution analyses (Figure 1k,l) indicate that the g-C_3_N_4_ (elements of C and N) compound covers completely and uniformly on the surfaces of {010}BiVO_4_ and {012}BiVO_4_ (elements of Bi, V, and O) manifesting the successful combinations between g-C_3_N_4_ and {010}BiVO_4_ or {012}BiVO_4_.

The successful fabrications of {010}BiVO_4_/g-C_3_N_4_ and {012}BiVO_4_/g-C_3_N_4_ can also be verified by XRD patterns and FTIR spectra. As shown in Figure 2a, it is found that the {010}BiVO_4_ and {012}BiVO_4_ exhibited strong diffraction peaks at about 2*θ* = 29.0°, 30.7°, 34.7°, 35.4°, 40.0°, 42.5°, 47.0°, 50.1°, 53.4°, 58.5°, and 59.7°, which corresponds to the (112), (004), (200), (020), (121), (015), (204), (220), (116), (312), and (026) crystal facets of the monoclinic BiVO_4_ (PDF#75-1867), respectively. At the same time, no other diffraction peak is involved indicating a pure crystal phase of BiVO_4_ [33]. After integration with g-C_3_N_4_, it is observed that no crystal destruction happened on both {010}BiVO_4_ and {012}BiVO_4_. No obvious g-C_3_N_4_ diffraction peaks appeared in both {010}BiVO_4_/0.2g-C_3_N_4_ and {012}BiVO_4_/0.1g-C_3_N_4_ samples, which may be ascribed to the small loading amount of g-C_3_N_4_ [34,35]. The structural composition of {010}BiVO_4_/0.2g-C_3_N_4_ and {012}BiVO_4_/0.1g-C_3_N_4_ were explored by FTIR spectra. As illustrated in Figure 2b, {010}BiVO_4_ appears the vibrational peaks at 470 and 612 cm^−1^, which should be attributed to the symmetric bending vibration of VO_4_^3−^ and the stretching vibration of Bi-O, respectively [36,37]. While for {012}BiVO_4_, it is observed that the peak wave numbers of symmetric bending vibration of VO_4_^3−^ and the stretching vibration of Bi-O decrease to 452.22 and 561.22 cm^−1^, respectively. The different vibrations of {010}BiVO_4_ and {012}BiVO_4_ may be attributed to the diversely exposed crystal planes. Additionally, the vibrational peaks of g-C_3_N_4_ from the {010}BiVO_4_/0.2g-C_3_N_4_ and {012}BiVO_4_/0.1g-C_3_N_4_ samples can also be detected at 1231, 1309, and 1399 cm^−1^, which are attributed to the C-N aromatic ring vibrations [30]. These confirm the successful integration between BiVO_4_ and g-C_3_N_4_.

The surface chemical compositions and electron states of {010}BiVO_4_/0.2g-C_3_N_4_ and {012}BiVO_4_/0.1g-C_3_N_4_ were analyzed by XPS. The survey XPS spectra (Figure 3a) demonstrate that the elements of Bi, V, and O existed in all of {010}BiVO_4_, {010}BiVO_4_/0.2g-C_3_N_4_, {012}BiVO_4_, and {012}BiVO_4_/0.1g-C_3_N_4_ samples, indicating the successful fabrications of BiVO_4_. The N peaks in the {010}BiVO_4_/0.2g-C_3_N_4_ and {012}BiVO_4_/0.1g-C_3_N_4_ were also detected at a binding energy of about 400 eV after loading g-C_3_N_4_, suggesting that g-C_3_N_4_ has been successfully loaded on both {010}BiVO_4_ and {012}BiVO_4_ surfaces. Moreover, the high-resolution XPS patterns of Bi 4f on {010}BiVO_4_, {010}BiVO_4_/0.2g-C_3_N_4_, {012}BiVO_4_, and {012}BiVO_4_/0.1g-C_3_N_4_ has been investigated. As shown in Figure 3b, peaks at 159.3 and 164.6 eV correspond to Bi^3+^ 4f_7/2_ and Bi^3+^ 4f_5/2_ [38], respectively, which implies that element Bi is present in {010}BiVO_4_ at +3 valence. While after introducing the g-C_3_N_4_ substrate, the locations of Bi^3+^ 4f_7/2_ and Bi^3+^ 4f_5/2_ shift to 159.1 and 164.4 eV, respectively. Similar results also have happened between {012}BiVO_4_ and {012}BiVO_4_/0.1g-C_3_N_4_, where the locations of Bi^3+^ 4f_7/2_ and Bi^3+^ 4f_5/2_ change from 159.1 and 164.4 eV to 159.0 and 164.3 eV, respectively. These indicate that the electrons transfer from BiVO_4_ to g-C_3_N_4_. The same result can also be reflected on the high-resolution XPS spectra of V 2p, as shown in Appendix A. Furthermore, The high-resolution XPS spectra of O 1s were illustrated with two or three peaks of deconvolution in Appendix A. Peaks at about 530.0 and 532.0 eV are assigned to the lattice O^2-^ and adsorbed O^2-^ molecules, respectively [39,40]. Interestingly, for {010}BiVO_4_, a characteristic peak of adsorbed O^2-^ was observed at about 531.4 eV, which can be attributed to the adsorbed -OH [41]. Instead, for {012}BiVO_4_, the characteristic peak of adsorbed O^2-^ is at about 532.5 eV that originates from the C=O adsorption [37]. These different properties can ascribe to the diversely exposed crystal planes, which are in accordance with the analysis of FTIR. Furthermore, after introducing g-C_3_N_4_ substrate, the binding energy of lattice O^2-^ decreased in both {010}BiVO_4_/0.2g-C_3_N_4_ and {012}BiVO_4_/0.1g-C_3_N_4_, indicating the electrons transfer from BiVO_4_ to g-C_3_N_4_ and corresponding to results of XPS spectra of Bi 4f and V 2p.

### 3.2. Construction of Energy Band Structure

To judge the heterojunction structures of {010}BiVO_4_/g-C_3_N_4_ and {012}BiVO_4_/g-C_3_N_4_, the energy band structures with alignments of {010}BiVO_4_, {012}BiVO_4_, and g-C_3_N_4_ should be tested and established. UV-diffuse absorption spectra of g-C_3_N_4_, {010}BiVO_4_, {012}BiVO_4_, {010}BiVO_4_/0.2g-C_3_N_4_, and {012}BiVO_4_/0.1g-C_3_N_4_ samples were first studied (Appendix A) and demonstrated decently visible absorptions maximum to 550 nm facilitating the subsequent visible light degradation performances. Based on the tauc equation, the bandgap values of {010}BiVO_4_, {012}BiVO_4_, and g-C_3_N_4_ were calculated to be 2.41, 2.39, and 2.80 eV (Figure 4a), which were transformed from the corresponding UV-diffuse absorption spectrum. The conduction band potentials of {010}BiVO_4_, {012}BiVO_4_, and g-C_3_N_4_ were calculated by electrochemical experiments. As shown in Figure 4b, positive tangents in Mott–Schottky plots of {010}BiVO_4_, {012}BiVO_4_, and g-C_3_N_4_ were observed, which reveals the n-type semiconductor characteristics of these prepared samples [42,43]. For n-type semiconductors, the conduction band (E_CB_) is considered very close to the flat band [44,45]. Hence, by calculating the intercept of the tangent line in Mott–Schottky plots, the conduction band energy potentials of {010}BiVO_4_, {012}BiVO_4_, and g-C_3_N_4_ were estimated as −0.45, −0.24 and −0.07 V vs. NHE at pH = 0, respectively. Since the bandgap energy has been confirmed by UV-diffuse absorption spectrum, through Formula (2), the valence band (E_VB_) potentials of {010}BiVO_4_, {012}BiVO_4_, and g-C_3_N_4_ were calculated to be 1.96, 2.15, and 2.73 V vs. NHE at pH = 0, respectively.
E_g_ = E_VB_ − E_CB_(2)

Matching the band energy of {010}BiVO_4_ and {012}BiVO_4_ with a g-C_3_N_4_ display that two types of heterojunctions can be formed for {010}BiVO_4_/g-C_3_N_4_ and {012}BiVO_4_/g-C_3_N_4_: type II heterojunction (Figure 4c) and Z-scheme heterojunction (Figure 4d). For type II heterojunction, under light irradiation, the photoinduced electrons from the conduction bands of {010}BiVO_4_ and {012}BiVO_4_ can flow to g-C_3_N_4_ with lower conduction band energy. Moreover, the photoinduced holes from the valence bands of g-C_3_N_4_ can inject into {010}BiVO_4_ and {012}BiVO_4_ with higher valence band energy potentials. Thus, theoretically, the photoelectrons accumulated on the conduction band of g-C_3_N_4_ and the photoholes on the valence bands of {010}BiVO_4_ and {012}BiVO_4_ can separately engage in the reduction and oxidation reactions. However, considering that both {010}BiVO_4_ and {012}BiVO_4_ have been wrapped by g-C_3_N_4_ sheets, merely the photoelectrons from the conduction band of g-C_3_N_4_ can participate in the anticipated reduction reaction. At the same time, the photoholes can only be trapped in {010}BiVO_4_ and {012}BiVO_4_ crystals. For Z-scheme heterojunction, under light irradiation, the photoinduced holes from the valence bands of {010}BiVO_4_ and {012}BiVO_4_ will integrate and dissipate with the photoinduced electrons from the conduction bands of g-C_3_N_4_, leaving the photoelectrons on the conduction bands of {010}BiVO_4_ and {012}BiVO_4_ as well as the photoholes on the valence band of g-C_3_N_4_. Similarly, given the morphologies of {010}BiVO_4_/g-C_3_N_4_ and {012}BiVO_4_/g-C_3_N_4_, only the photoholes on the valence band of g-C_3_N_4_ can engage in the desirable oxidation reaction. Based on the above analyses of carrier transfer, the actual heterojunctions of {010}BiVO_4_/g-C_3_N_4_ and {012}BiVO_4_/g-C_3_N_4_, either type II heterojunction or Z-scheme heterojunction, can be figured out by radical tests and photodegradation performances interpretations.

### 3.3. Radical Species Analyses

To figure out the heterojunction types of {010}BiVO_4_/g-C_3_N_4_ and {012}BiVO_4_/g-C_3_N_4_, the radical species have been analyzed. According to the previous reports [46,47], the ^•^OH radical generation potential from H_2_O oxidation is 2.38 V vs. NHE at pH = 0 (Equation (3)), and O_2_ can be reduced to O_2_^•−^ radical by photoelectrons at −0.046 V vs. NHE at pH = 0 (Equation (4)).
H_2_O + h^+^ → ^•^OH + H^+^      E = 2.38 V vs. NHE at pH = 0(3)
O_2_ + e^−^ → O_2_^•−^      E = −0.046 vs. NHE at pH = 0(4)

Therefore, based on the energy band structures, for {010}BiVO_4_/g-C_3_N_4_ and {012}BiVO_4_/g-C_3_N_4_ with type II heterojunction (Figure 4c), the O_2_^•−^ should be the main radical species, since the photoelectrons of {010}BiVO_4_ and {012}BiVO_4_ will flow into the g-C_3_N_4_ shell for O_2_ reduction while the photoholes of g-C_3_N_4_ will be trapped in the inner core of {010}BiVO_4_ and {012}BiVO_4_. On the contrary, for Z-scheme heterojunction (Figure 4d), the ^•^OH should be the dominant radical species because the photoholes of g-C_3_N_4_ can oxidize H_2_O into ^•^OH, but the photoelectrons are arrested by inner {010}BiVO_4_ and {012}BiVO_4_.

To reveal the practical radical species fabrications, the EPR spectra were carried out for g-C_3_N_4_, {010}BiVO_4_/0.2g-C_3_N_4_ and {012}BiVO_4_/0.1g-C_3_N_4_ systems under aerobic conditions. Both ^•^OH (Figure 5a) and O_2_^•−^ radicals (Figure 5b) were generated using g-C_3_N_4_ as a photocatalyst, which is reasonable as the valence band and conduction band energy potentials of g-C_3_N_4_ are more positive and negative than H_2_O oxidation and O_2_ reduction potentials. After the combination of {010}BiVO_4_ or {012}BiVO_4_ with g-C_3_N_4_, the concentrations of ^•^OH radical decreased considerably, while the concentration promotions of O_2_^•−^ radical were observed. The change trends of ^•^OH and O_2_^•−^ radical conform to the type II heterojunction. Therein, the photoelectrons from {010}BiVO_4_ or {012}BiVO_4_ transfer into g-C_3_N_4,_ elevating the O_2_ reduction for O_2_^•−^ radical fabrication. However, the photoholes of g-C_3_N_4_ migrate to the wrapped {010}BiVO_4_ or {012}BiVO_4_ inhibiting the H_2_O oxidation into ^•^OH radical. Therefore, both heterojunctions of {010}BiVO_4_/g-C_3_N_4_ and {012}BiVO_4_/g-C_3_N_4 have_ proven to be type II heterojunctions.

### 3.4. Photocatalytic Performances

The type II heterojunctions of {010}BiVO_4_/g-C_3_N_4_ and {012}BiVO_4_/g-C_3_N_4_ can also be proven by their photocatalytic performances. The photocatalytic performances of {010}BiVO_4_, {010}BiVO_4_/g-C_3_N_4_, {012}BiVO_4_, and {012}BiVO_4_/g-C_3_N_4_ were investigated by degradations of TC under visible light irradiation. Compared with pure {010}BiVO_4_ and {012}BiVO_4_, the additions of g-C_3_N_4_ evidently promote the degradation performances of TC molecules. As shown in Figure 6a, after 60 min of light exposure, a maximum of 56% of TC was degraded by {010}BiVO_4_/0.2g-C_3_N_4_, while only 26% was degraded by {012}BiVO_4_/0.1g-C_3_N_4_. These different degradation performances can be explained by the type II heterogeneous structure, as the {010}BiVO_4_ equips a higher conduction band than {012}BiVO_4_ driving more photoelectrons into g-C_3_N_4_ and promoting a larger amount of O_2_^•−^ radicals for TC degradation. Moreover, the Z-scheme heterojunction cannot interpret the performance variation. Because, if it is the Z-scheme heterojunction, equal photoholes should remain on the valence band of g-C_3_N_4,_ producing the same amount of ^•^OH radical and resulting in the same TC degradation performances between {010}BiVO_4_/g-C_3_N_4_ and {012}BiVO_4_/g-C_3_N_4_. Thus, the Z-scheme heterojunction is repelled. Subsequently, as shown in Figure 6a, the further loading of g-C_3_N_4_ led to the reduced degradation performances on both {010}BiVO_4_/0.3g-C_3_N_4_ and {012}BiVO_4_/0.3g-C_3_N_4_, which may be attributed to the thick g-C_3_N_4_ shells inhibiting the light absorption of BiVO_4_. Thus no sufficient photoelectrons were produced and transferred from BiVO_4_ to g-C_3_N_4_ to guarantee the high degradation performances. The kinetic analyses of these photocatalytic reactions were further explored, and the results suggested that they all conformed to the first-order reaction kinetic equations (Figure 6b). The corresponding kinetic rate constant of {010}BiVO_4_ was estimated to be 0.0064 min^−1^ (Figure 6c), and with the continuous g-C_3_N_4_ loading, the kinetic rate constant of {010}BiVO_4_/0.2g-C_3_N_4_ increased to be 0.0130 min^−1^. While for {012}BiVO_4_, the kinetic rate constant only achieved 0.0037 min^−1^ and increased to 0.0051 min^−1^ for {012}BiVO_4_/0.1g-C_3_N_4_. Furthermore, the cycling experiments showed that after four cycles (Figure 6d), the degradation efficiency of {010}BiVO_4_/g-C_3_N_4_ exhibited merely a 3% reduction, which represents a relatively stable performance in the field of photocatalysis [48,49,50]. The minor decline in degradation efficiency may be ascribed to the attenuation of surface adsorption with recycling. As shown in Appendix A, the BET spectrum indicates that the specific surface area of {010}BiVO_4_/g-C_3_N_4_ is about 9.6816 m^2^/g, which represents excellent adsorption performance and can conduce to the photocatalytic activity [51]. However, during the photodegradation of TC, some macromolecular substances should be inevitably absorbed on the surface of {010}BiVO_4_/g-C_3_N_4_ ((Figure 7b) and affect the photocatalytic activity, which could not be emancipate by water clean during the recycle experiment. Therefore, a light degradation efficiency reduction emerged with the recycling of photocatalysis [52].

### 3.5. Photocatalytic Degradation Mechanism

To thoroughly and exhaustively explain the mechanism, the roles of radicals were tested. Taking benzoquinone (BQ), isopropyl alcohol (IPA), and triethanolamine (TEOA) as O_2_^•−^, ^•^OH, and photoholes capturers, the photocatalytic degradation of TC were performed. As shown in Figure 7a, the TC degradation by {010}BiVO_4_/0.2g-C_3_N_4_ was reduced from 56% to 48% when O_2_^•−^ trapping agent BQ was introduced. Moreover, the degradation rate was almost unchanged by the addition of ^•^OH trapping agent IPA. This indicates that O_2_^•−^ rather than ^•^OH is the active radical species for TC degradation over BiVO_4_/g-C_3_N_4_ photocatalyst. The introduction of the hole-trapping agent of TEOA, the degradation rate increased to 83%. This is because TEOA could trap a portion of the holes, inhibiting the recombination of photogenerated carriers, thus leaving more electrons to participate in O_2_^•−^ generation for TC degradation.

The TC degradation process was speculated based on the degradation intermediates detection by LC-MS. As shown in Figure 7b and Appendix A, the degradation intermediates of TC over {010}BiVO_4_, {010}BiVO_4_/0.2g-C_3_N_4_, {012}BiVO_4_, and {012}BiVO_4_/0.1g-C_3_N_4_ are almost identical. Through the molecular debris detection with different m/z values, we speculate that O_2_^•−^ first promotes the dehydroxylation and demethylation of TC to produce the products with m/z values of 410 and 427. Subsequently, they continued to undergo catalytic ring-opening reactions, decomposing into small molecules with an m/z value of 186, which were eventually broken down completely into CO_2_ and H_2_O.

Combined with the band structure, heterojunction type, free radical type, and TC degradation intermediates, the photocatalytic mechanisms on {010}BiVO_4_/g-C_3_N_4_ and {012}BiVO_4_/g-C_3_N_4_ can be speculated (Figure 8). Through the band combination between {010}BiVO_4_ or {012}BiVO_4_ with g-C_3_N_4_, the type II heterojunctions can be formed. Under visible light irradiation, the photoelectrons generated from {010}BiVO_4_ and {012}BiVO_4_ can both flow into the conduction band of g-C_3_N_4_ to engage the formation of O_2_^•−^ radical. While the photoholes from g-C_3_N_4_ ingrate into the valence bands of {010}BiVO_4_ and {012}BiVO_4_. Since {010}BiVO_4_ and {012}BiVO_4_ are encased in g-C_3_N_4_ sheets, the trapped photoholes are hard to participate in some reactions. Therefore, O_2_^•−^ became the main active radical for TC degradation in both {010}BiVO_4_/g-C_3_N_4_ and {012}BiVO_4_/g-C_3_N_4_ systems. Finally, the TC molecules go through dihydroxylation, demethylation, and ring-opening, being decomposed into CO_2_ and H_2_O molecules.

## 4. Conclusions

In summary, to figure out the confusing heterojunction type between BiVO_4_ and g-C_3_N_4_, we rationally designed {010}BiVO_4_/g-C_3_N_4_ and {012}BiVO_4_/g-C_3_N_4_ as core-shell structures, allowing only photoelectrons or photoelectrodes to participate in oxidation or reduction reactions. Through energy band establishment, radical species investigation, photocatalytic TC degradation performances, and LC-MS tests, both {010}BiVO_4_/g-C_3_N_4_ and {012}BiVO_4_/g-C_3_N_4_ exhibit the type II heterojunction structures rather than the previously reported Z-scheme heterojunctions. The O_2_^•−^ was proven as the main active radical for TC decomposition in both {010}BiVO_4_/g-C_3_N_4_ and {012}BiVO_4_/g-C_3_N_4_ systems. Moreover, owing to the more negative conduction band of {010}BiVO4 compared with {012}BiVO_4_, the {010}BiVO_4_/g-C_3_N_4_ demonstrated higher TC degradation efficiency than {012}BiVO_4_/g-C_3_N_4_. Finally, the photocatalytic mechanism of TC degradation was proposed based on the band structure, heterojunction type, free radical changes, and TC degradation intermediates. We expect that this work can become a reference for the construction of heterogeneous complexes.

## Figures and Tables

**Figure 1 ijerph-19-14770-f001:**
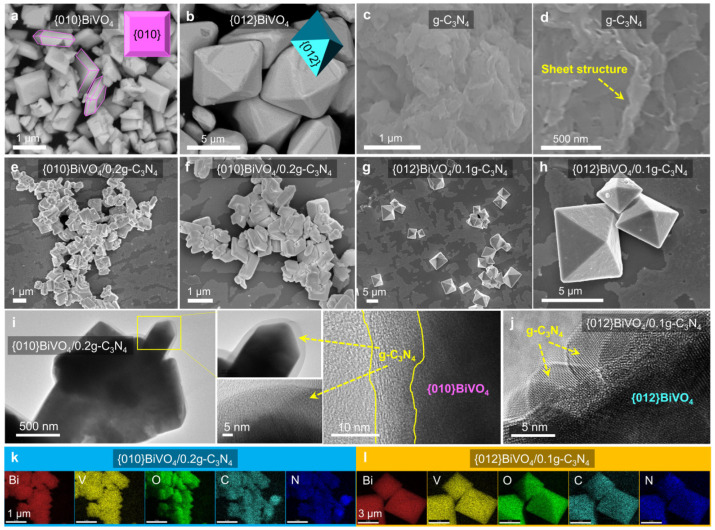
(**a**) SEM images of {010}BiVO_4_, (**b**) {012}BiVO_4_, (**c**,**d**) g-C_3_N_4_, (**e**,**f**) {010}BiVO_4_/0.2g-C_3_N_4_, and (**g**,**h**) {012}BiVO_4_/0.1g-C_3_N_4_. (**i**) TEM images of {010}BiVO_4_/0.2g-C_3_N_4_ and (**j**) {012}BiVO_4_/0.1g-C_3_N_4_. (**k**) SEM mapping images of {010}BiVO_4_/0.2g-C_3_N_4_ and (**l**) {012}BiVO_4_/0.1g-C_3_N_4_.

**Figure 2 ijerph-19-14770-f002:**
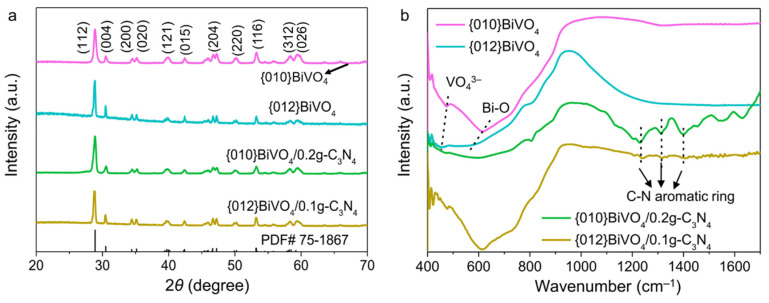
(**a**) XRD patterns and (**b**) FTIR spectra of {010}BiVO_4_, {012}BiVO_4_, {010}BiVO_4_/0.2g-C_3_N_4_, and {012}BiVO_4_/0.1g-C_3_N_4_.

**Figure 3 ijerph-19-14770-f003:**
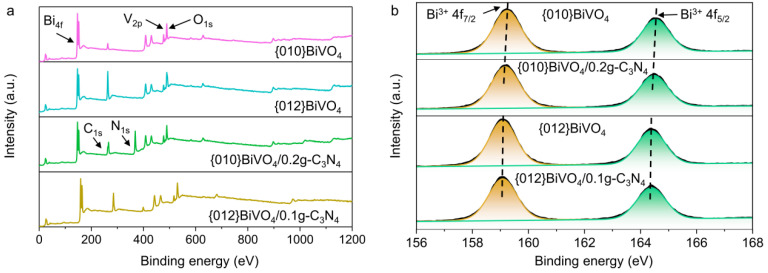
(**a**) Survey XPS spectra and (**b**) high-resolution XPS spectra of Bi 4f on {010}BiVO_4_, {010}BiVO_4_/0.2g-C_3_N_4_, {012}BiVO_4_, and {012}BiVO_4_/0.1g-C_3_N_4_.

**Figure 4 ijerph-19-14770-f004:**
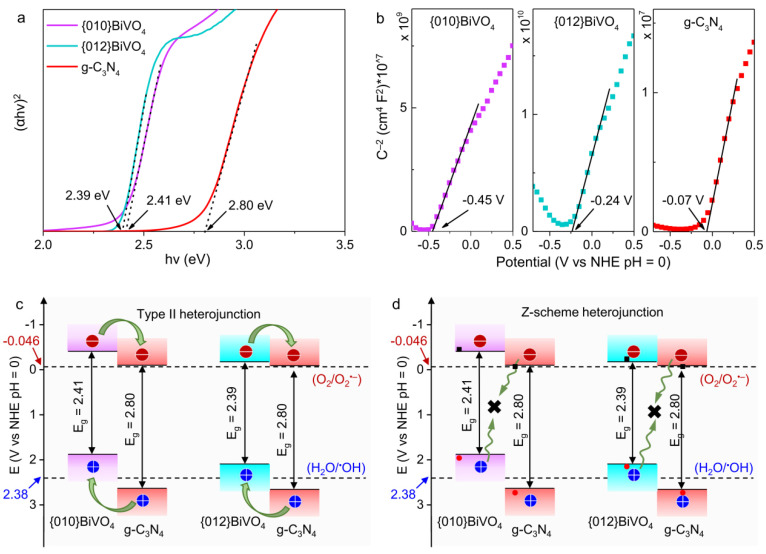
(**a**) Tauc curves of {010}BiVO_4_, {012}BiVO_4_, and g-C_3_N_4_ for bandgap tests. (**b**) Mott–Schottky plots of {010}BiVO_4_, {012}BiVO_4_, and g-C_3_N_4_ for conduction band energy tests. (**c**) The band structures of {010}BiVO_4_/g-C_3_N_4_ and {012}BiVO_4_/g-C_3_N_4_ with type II heterojunction. (**d**) The band structures of {010}BiVO_4_/g-C_3_N_4_ and {012}BiVO_4_/g-C_3_N_4_ with Z-scheme heterojunction.

**Figure 5 ijerph-19-14770-f005:**
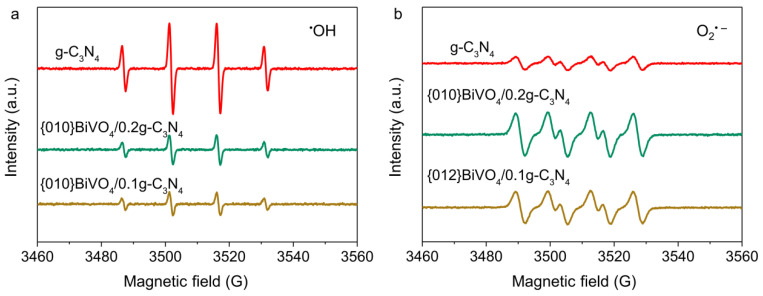
(**a**) EPR spectra of ^•^OH and (**b**) O_2_^•−^ radicals for g-C_3_N_4_, {010}BiVO_4_/0.2g-C_3_N_4_, and {012}BiVO_4_/0.1g-C_3_N_4_.

**Figure 6 ijerph-19-14770-f006:**
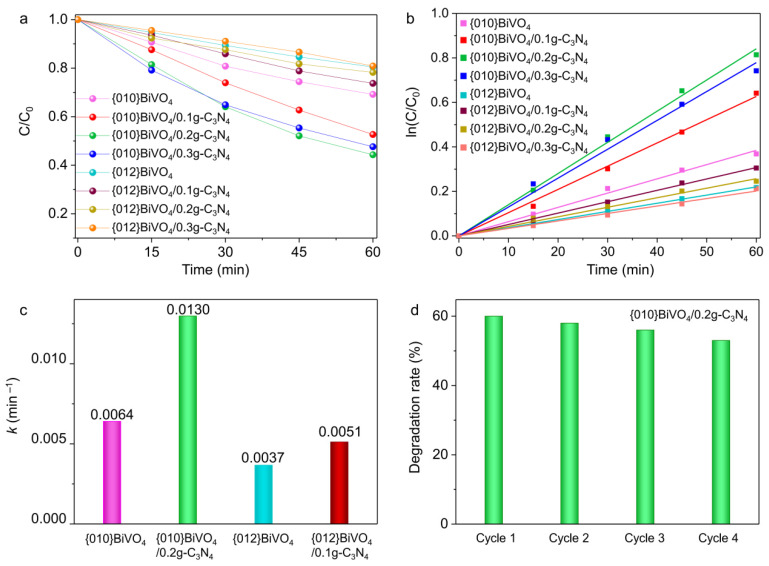
(**a**) The photocatalytic degradation curves of TC on {010}BiVO_4_ and {012}BiVO_4_ with different g-C_3_N_4_ loading amounts. (**b**) The kinetic analyses of TC degradation curves with (**c**) the degradation rate constant. (**d**) Stability tests of TC degradation over {010}BiVO_4_/g-C_3_N_4_.

**Figure 7 ijerph-19-14770-f007:**
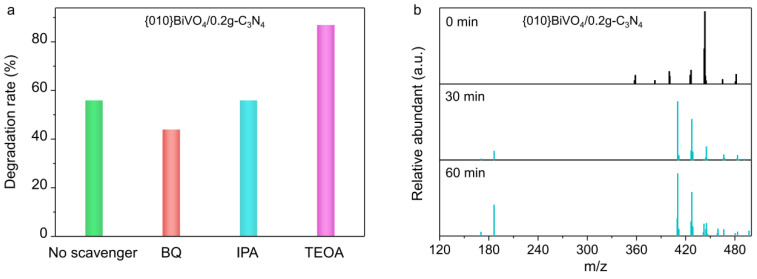
(**a**) Trapping experiments of active species during the photocatalytic degradation of TC over {010}BiVO_4_/0.2g-C_3_N_4_. (**b**) LC-MS spectra of intermediates over {010}BiVO_4_/0.2g-C_3_N_4_ during TC degradation.

**Figure 8 ijerph-19-14770-f008:**
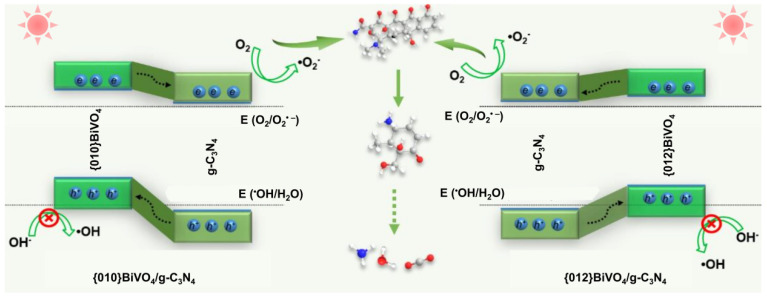
Photocatalytic mechanism speculations over {010}BiVO_4_/g-C_3_N_4_ and {012}BiVO_4_/g-C_3_N_4_.

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
