# Peer review of "Type II Heterojunction Formed between {010} or {012} Facets Dominated Bismuth Vanadium Oxide and Carbon Nitride to Enhance the Photocatalytic Degradation of Tetracycline"

_ijerph, 2022, doi:10.3390/ijerph192214770_

Round 1

Reviewer 1 Report

The manuscript entitled “Type Ⅱ heterojunction formed between {010} or {012} facets dominated bismuth vanadium oxide and carbon nitride to enhance the photocatalytic degradation of tetracycline” by Zhang and coauthor report, {010}C and {012} facets dominated bismuth vanadium oxide were rationally fabricated and decorated by carbon nitride to investigate their heterojunction structure, simultaneously the corresponding photocatalytic efficiencies were tested. Two disparate exposed crystal planes of bismuth vanadate were used to investigate the effect of crystal planes on the type of heterojunction. And the core-shell structure, that is bismuth vanadate encased in carbon nitride, was created to make sure the same photocatalytic reaction interface. Through energy band establishment, radical species investigation, both {010} and {012} facets dominated bismuth vanadium oxide/carbon nitride composites exhibit the type II heterojunction structures rather than the Z-scheme heterojunctions. Furthermore, to investigate the effect of type II heterojunction, the photocatalytic tetracycline degradations were performed finding that {010} facets dominated bismuth vanadium oxide/carbon nitride composite demonstrated the higher degradation efficiencies than that of {012} facets, due to the higher conduction band energy. Besides, through the free radical trapping experiments and intermediate detections of degradation products, the superoxide radical was proven to be the main active radical to decompose the tetracycline molecules. Therein, the tetracycline molecules were degraded via dihydroxylation-demethylation-ring opening reactions into water and carbon dioxide. This work is anticipated to provide some basic research and theoretical supports for the progressive and controlled synthesis of photocatalysts with heterojunction structures. However, I have some concerns, and this work is not publishable in IJERPH.

1.       The title, abstract & conclusion of this manuscript seems to be very long and confusing. The author needs to summarize in a better way that better reflects the theme and idea of the paper.

2.       Figure 1 is confusing. The author needs to show only the representative data throught the main manuscript that better reflects the main idea of work and put all other data in supporting information.

3.       The author needs to improve the English of this paper with a professional or with a native English speaker.

4.       I think this paper is also beyond the scope of IJERPH.

Author Response

Question 1: The title, abstract & conclusion of this manuscript seems to be very long and confusing. The author needs to summarize in a better way that better reflects the theme and idea of the paper.

Our response: We are grateful for your constructive comment that would greatly improve the quality of our work. We have revised the abstract and conclusions, and the results are as follows:

Abstract: Both type II and Z schemes can explain the charge transfer behavior of the heterojunction structure well, but the type of heterojunction structure formed between bismuth vanadium oxide and carbon nitride still have not been clarified. Herein, we rationally prepared bismuth vanadium oxide with {010} and {012} faces predominantly, and carbon nitride as a decoration to construct a core-shell structure with bismuth vanadium oxide wrapped in carbon nitride to ensure the same photocatalytic reaction interface. Through energy band establishment, radical species investigation, both {010} and {012} facets dominated bismuth vanadium oxide/carbon nitride composites exhibit the type II heterojunction structures rather than the Z-scheme heterojunctions. Furthermore, to investigate the effect of type II heterojunction, the photocatalytic tetracycline degradations were performed finding that {010} facets dominated bismuth vanadium oxide/carbon nitride composite demonstrated the higher degradation efficiency than that of {012} facets, due to the higher conduction band energy. Besides, through the free radical trapping experiments and intermediate detection of degradation products, the superoxide radical proven to be the main active radical to decompose the tetracycline molecules. Therein, the tetracycline molecules were degraded to water and carbon dioxide by dihydroxylation-demethylation-ring opening reactions. This work investigates the effect of crystal planes on heterojunction types through two different exposed crystal planes of bismuth vanadate oxide, which can provide some basic research and theoretical support for the progressive and controlled synthesis of photocatalysts with heterojunction structures………………In summary, to figure out the confusing heterojunction type between BiVO4 and g-C3N4, we rationally designed {010}BiVO4/g-C3N4 and {012}BiVO4/g-C3N4 as core-shell structures, allowing only photoelectrons or photoelectrodes to participate in oxidation or reduction reactions. Through energy band establishment, radical species investigation, photocatalytic TC degradation performances and LC-MS tests,both {010}BiVO4/g-C3N4 and {012}BiVO4/g-C3N4 exhibit the type II heterojunction structures rather the previously reported Z-scheme heterojunctions. Besides, the O2•– proven as the main active radical for TC decomposition in both {010}BiVO4/g-C3N4 and {012}BiVO4/g-C3N4 system. Moreover, owing to the more negative conduction band of {010}BiVO4 comparing with {012}BiVO4, the {010}BiVO4/g-C3N4 demonstrated higher TC degradation efficiency than {012}BiVO4/g-C3N4. Finally, the photocatalytic mechanism of TC degradation was proposed based on the band structure, heterojunction type, free radical changes and TC degradation intermediates. We expect that this work can become a reference for the construction of heterogeneous complexes.

Question 2: Figure 1 is confusing. The author needs to show only the representative data throught the main manuscript that better reflects the main idea of work and put all other data in supporting information.

Our response: Thanks a lot for your valuable suggestion. We greatly identify with the view that only the representative data throught the main manuscript that better reflects the main idea of work and all other data in should be listed in supporting information. Figure R1 describe the SEM and TEM images of {010}BiVO4, {012}BiVO4, g-C3N4, {010}BiVO4/g-C3N4 and {012}BiVO4/g-C3N4, which distinctly reflect the microstructures of these samples and also are essential and intuitive evidence for the combine between g-C3N4 and BiVO4. Figure R1a represent the pure {010}BiVO4 is sheet-shaped with a length of ca. 1 μm, while Figure R1b depicts the {012}BiVO4 is near-octahedral shaped with the size of about 5 μm. Figure R1c and R1d exhibit the sheet structure morphology of g-C3N4. Figure R1e-1h are important evidences for the successful loading of g-C3N4 onto BiVO4, which also suggests {010}BiVO4/0.2g-C3N4 and {012}BiVO4/0.1g-C3N4 retained the original flake or octahedral shapes and unchanged size. Intuitively, Figure R1i-1j prove nanosheet shell of g-C3N4 covered on the surface of BiVO4, which also simultaneously display the microstructure of these samples. Moreover, Figure R1k and R1l draw the chemical element distribution of C, N, Bi, V and O, which is an essential and visual proof for that g-C3N4 (elements of C and N) compound covers completely and uniformly on the surfaces of {010}BiVO4 and {012}BiVO4 (elements of Bi, V and O) manifesting the successful combinations between g-C3N4 and {010}BiVO4 or {012}BiVO4. These results are conventionally listed in the main manuscript [1-3]. Meanwhile, as your view, other data such as high resolution XPS spectra of V 2p and O1s, UV-Vis diffuse reflectance spectra and LC-MS spectra of intermediates over {010}BiVO4, {012}BiVO4 and {012}BiVO4/0.1g-C3N4 during TC degradation are putted in the supporting information.

Figure R1. (a) SEM images of {010}BiVO4, (b) {012}BiVO4, (c, d) g-C3N4,(e,f) {010}BiVO4/0.2g-C3N4 and (g, h) {012}BiVO4/0.1g-C3N4. (i) TEM images of {010}BiVO4/0.2g-C3N4 and (j) {012}BiVO4/0.1g-C3N4. (k) SEM mapping images of {010}BiVO4/0.2g-C3N4 and (l) {012}BiVO4/0.1g-C3N4. (Figure R1 corresponds to Fig. 1 in the revised manucript)

Ref:

[1] Y. Wang, G.Q. Tan, T. Liu, Y.N. Su, H.J. Ren, X.L. Zhang, A. Xia, L. Lv, Y. Liu, Photocatalytic properties of the g-C3N4/{010} facets BiVO4 interface Z-Scheme photocatalysts induced by BiVO4 surface heterojunction, Appl. Catal. B-Environ., 234 (2018) 37-49.

[2] M.Y. Dang, G.Q. Tan, M. Wang, B.X. Zhang, Y. Wang, L. Lv, H.J. Ren, A. Xia, Enhanced photocatalytic performance of g-C3N4-BiVO4-Ag heterojunction induced by interfacial electric fields and Schottky junction, J. Alloy. Compd., 897 (2022) 11.

[3] M. Ou, S.P. Wan, Q. Zhong, S.L. Zhang, Y. Song, L.N. Guo, W. Cai, Y.L. Xu, Hierarchical Z-scheme photocatalyst of g-C3N4@Ag/BiVO4 (040) with enhanced visible-light-induced photocatalytic oxidation performance, Appl. Catal. B-Environ., 221 (2018) 97-107.

Question 3: The author needs to improve the English of this paper with a professional or with a native English speaker.

Our response: Thanks for your valuable suggestion. The manuscript and supporting information both have been carefully proofread with the help of a native English speaker in order to make the article as comprehensible as possible. And all the corrections have been marked in red, which are hoped to meet your approval.

Question 4: I think this paper is also beyond the scope of IJERPH.

Our response: Thanks a lot for your view. IJERPH covers environmental sciences and engineering, public health, environmental health, occupational hygiene, health economic and global health research, etc. This work should belong to environmental sciences and engineering or environmental health, which is under the scope of IJERPH. Meanwhile, this work was submitted as an article for the special issue “Photocatalysis Assists Carbon Neutrality” what is a special issue focused on the research for photocatalysis in the field of environmental. Obviously, this work accurately match the topic of the special issue. Moreover, special issue “Photocatalysis Assists Carbon Neutrality” has been published plentiful studies in these field [4-15]. This work distinctively investigates the effect of crystal planes on heterojunction types through two different exposed crystal planes of bismuth vanadate oxide, which can provide some basic research and theoretical support for the progressive and controlled synthesis of photocatalysts with heterojunction structures. It might be contribute to reinforce the research achievement of IJERPH in the field of environmental sciences and engineering or environmental health.

Ref:

[4] X. Xu, F.Y. Ji, Z.H. Fan, L. He, Degradation of Glyphosate in Soil Photocatalyzed by Fe3O4/SiO2/TiO2 under Solar Light, Int. J. Environ. Res. Public Health, 8 (2011) 1258-1270.

[5] X.P. Luo, C.F. Chen, J. Yang, J.Y. Wang, Q. Yan, H.Q. Shi, C.Y. Wang, Characterization of La/Fe/TiO2 and Its Photocatalytic Performance in Ammonia Nitrogen Wastewater, Int. J. Environ. Res. Public Health, 12 (2015) 14626-14639.

[6] W.C. Liao, V.K. Sharma, S. Xu, Q.S. Li, L. Wang, Microwave-Enhanced Photolysis of Norfloxacin: Kinetics, Matrix Effects, and Degradation Pathways, Int. J. Environ. Res. Public Health, 14 (2017) 17.

[7] H.W. Huo, X.J. Hu, H. Wang, J. Li, G.Y. Xie, X.F. Tan, Q. Jin, D.X. Zhou, C. Li, G.Q. Qiu, Y.G. Liu, Synergy of Photocatalysis and Adsorption for Simultaneous Removal of Hexavalent Chromium and Methylene Blue by g-C3N4/BiFeO3/Carbon Nanotubes Ternary Composites, Int. J. Environ. Res. Public Health, 16 (2019) 18.

[8] G.L. Li, Z.G. Hou, R.H. Zhang, X.L. Chen, Z.B. Lu, Nanometer Titanium Dioxide Mediated High Efficiency Photodegradation of Fluazifop-p-Butyl, Int. J. Environ. Res. Public Health, 16 (2019) 11.

[9] F.Y. Li, M.X. Lin, Synthesis of Biochar-Supported K-doped g-C3N4 Photocatalyst for Enhancing the Polycyclic Aromatic Hydrocarbon Degradation Activity, Int. J. Environ. Res. Public Health, 17 (2020) 15.

[10] O.C. Olatunde, D.C. Onwudiwe, Graphene-Based Composites as Catalysts for the Degradation of Pharmaceuticals, Int. J. Environ. Res. Public Health, 18 (2021) 36.

[11] N. Nasseh, R. Khosravi, N.S.M. Moghaddam, S. Rezania, Effect of UVC and UVA Photocatalytic Processes on Tetracycline Removal Using CuS-Coated Magnetic Activated Carbon Nanocomposite: A Comparative Study, Int. J. Environ. Res. Public Health, 18 (2021) 21.

[12] H. Liu, H.W. Chen, N. Ding, Visible Light-Based Ag3PO4/g-C3N4@MoS2 for Highly Efficient Degradation of 2-Amino-4-acetylaminoanisole (AMA) from Printing and Dyeing Wastewater, Int. J. Environ. Res. Public Health, 19 (2022) 14.

[13] Y.M. Yu, K. Liu, Y.Y. Zhang, X. Xing, H. Li, High Photocatalytic Activity of g-C3N4/La-N-TiO2 Composite with Nanoscale Heterojunctions for Degradation of Ciprofloxacin, Int. J. Environ. Res. Public Health, 19 (2022) 17.

[14] X. Guo, L. Rao, Z.Y. Shi, Preparation of High-Porosity B-TiO2/C3N4 Composite Materials: Adsorption-Degradation Capacity and Photo-Regeneration Properties, Int. J. Environ. Res. Public Health, 19 (2022) 16.

[15] J.Y. Zhu, Y.Y. Zhu, Z. Chen, S.J. Wu, X.J. Fang, Y. Yao, Progress in the Preparation and Modification of Zinc Ferrites Used for the Photocatalytic Degradation of Organic Pollutants, Int. J. Environ. Res. Public Health, 19 (2022) 32.

Reviewer 2 Report

1. Abstract should be aim and achievements 

2. Author should mention the model of the equipment

3. All XRD peaks should be indexed

4. All core-level XPS graphs of O1s should supply

5.  Photoelectrochemical analysis must supply in the revised MS

Author Response

Question 1: Abstract should be aim and achievements.

Our response: We are grateful for your constructive comment that would greatly improve the quality of our work. We have concised the abstract in the revised manuscript, and the result is as follows:

Abstract: Both type II and Z schemes can explain the charge transfer behavior of the heterojunction structure well, but the type of heterojunction structure formed between bismuth vanadium oxide and carbon nitride still have not been clarified. Herein, we rationally prepared bismuth vanadium oxide with {010} and {012} faces predominantly, and carbon nitride as a decoration to construct a core-shell structure with bismuth vanadium oxide wrapped in carbon nitride to ensure the same photocatalytic reaction interface. Through energy band establishment, radical species investigation, both {010} and {012} facets dominated bismuth vanadium oxide/carbon nitride composites exhibit the type II heterojunction structures rather than the Z-scheme heterojunctions. Furthermore, to investigate the effect of type II heterojunction, the photocatalytic tetracycline degradations were performed finding that {010} facets dominated bismuth vanadium oxide/carbon nitride composite demonstrated the higher degradation efficiency than that of {012} facets, due to the higher conduction band energy. Besides, through the free radical trapping experiments and intermediate detection of degradation products, the superoxide radical proven to be the main active radical to decompose the tetracycline molecules. Therein, the tetracycline molecules were degraded to water and carbon dioxide by dihydroxylation-demethylation-ring opening reactions. This work investigates the effect of crystal planes on heterojunction types through two different exposed crystal planes of bismuth vanadate oxide, which can provide some basic research and theoretical support for the progressive and controlled synthesis of photocatalysts with heterojunction structures.

Question 2: Author should mention the model of the equipment.

Our response: Thanks a lot for your careful reading and critical comment. The model of the equipment is supplied in the revised manuscript, and the result is as follows:

The crystal phase and purity of the catalyst were determined by X-ray diffractive apparatus (XRD, Palytical X'Pert Powder) of Panaco. The microstructure and elements of the powder samples were analyzed by field emission scanning electron microscopy (SEM, JSM-7001F) and energy dispersive X-ray spectroscopy (EDX, JSM-701F). The structural characteristics of the samples were observed through transmission electron microscope (TEM, JSM-2100F) made by Hitachi. The surface element composition and existence state of the synthesized materials were analyzed by X-ray photoelectron spectroscopy (XPS, Thermo Fisher Scientific K-Alpha). The chemical bonds and groups in the samples were analyzed and determined by fourier infrared spectroscopy (FT-IR, Thermo 6700) from Thermo Field Company. chemical bonds and groups in the samples were analyzed. The hydroxyl radical and superoxide radical of the samples were analyzed by electron paramagnetic resonance (EPR, A300) spectrometer of Brook Corporation. The comparative area of the samples was measured by the BET tester(BET, ASAP2460) of Mack Company. The intermediate products of the photocatalytic degradation of TC were tested by waters single quadrupole liquid mass spectrometry (LC-MS, ZQ2000). The optical absorption properties of the synthesized materials were characterized by UV-Vis diffuse reflectance spectroscopy (UV-VIS DRS, U-3900) from Hitachi, which scanning range was 200-800 nm, and BaSO4 powder was used as a reference.

Question 3: All XRD peaks should be indexed

Our response: Thanks a lot for your enlightened suggestion. As shown in Figure R2, diffraction peaks of {010}BiVO4 and {012}BiVO4 at about 2θ = 29.0°, 30.7°, 34.7°, 35.4°, 40.0°, 42.5°, 47.0°, 50.1°, 53.4°, 58.5° and 59.7° are correspond to the (112), (004), (200), (020), (121), (015), (204), (220), (116), (312) and (026) crystal facets of the monoclinic BiVO4, respectively.(PDF#75-1867).While, no other diffraction peak is involved indicating a pure crystal phase of BiVO4 [16]. After integration with g-C3N4, it is observed that no crystal destruction happened on both {010}BiVO4 and {012}BiVO4. No obvious g-C3N4 diffraction peaks appeared in both {010}BiVO4/0.2g-C3N4 and {012}BiVO4/0.1g-C3N4 samples might attributes to the small loading amount of g-C3N4 [17-22]which is further verified by SEM, TEM, FT-TR and XPS etc. The above analysis is also revised in the revised manuscript as “red” marked, and the result is as follows:

As shown in Fig. 2a, it is found that the {010}BiVO4 and {012}BiVO4 exhibited strong diffraction peaks at about 2θ = 29.0°, 30.7°, 34.7°, 35.4°, 40.0°, 42.5°, 47.0°, 50.1°, 53.4°, 58.5° and 59.7°, which corresponds to the (112), (004), (200), (020), (121), (015), (204), (220), (116), (312) and (026) crystal facets of the monoclinic BiVO4 (PDF#75-1867), respectively. While no other diffraction peak is involved indicating a pure crystal phase of BiVO4. After integration with g-C3N4, it is observed that no crystal destruction happened on both {010}BiVO4 and {012}BiVO4. No obvious g-C3N4 diffraction peaks appeared in both {010}BiVO4/0.2g-C3N4 and {012}BiVO4/0.1g-C3N4 samples, which may be ascribed to the small loading amount of g-C3N4.

Ref:

[16] P. Praus, J. Lang, A. Martaus, L. Svoboda, V. Matějka, M. Kormunda, M. Šihor, M. Reli, K. Kočí, Composites of BiVO4 and g-C3N4: synthesis, properties and photocatalytic decomposition of azo dye AO7 and nitrous oxide, J. Inorg. Organomet. Polym. Mater., 29 (2019) 1219-1234.

[17] M.Y. Dang, G.Q. Tan, M. Wang, B.X. Zhang, Y. Wang, L. Lv, H.J. Ren, A. Xia, Enhanced photocatalytic performance of g-C3N4-BiVO4 -Ag heterojunction induced by interfacial electric fields and Schottky junction, J. Alloy. Compd., 897 (2022) 11.

[18] M. Ou, S. P. Wan, Q. Zhong, S.L. Zhang, Y. Song, L.N. Guo, W. Cai, Y.L. Xu, Hierarchical Z-scheme photocatalyst of g-C3N4@Ag/BiVO4 (040) with enhanced visible-light-induced photocatalytic oxidation performance, Appl. Catal. B-Environ., 221 (2018) 97-107.

[19] M. Ou, Q. Zhong, S.L. Zhang, Synthesis and characterization of g-C3N4/BiVO4 composite photocatalysts with improved visible-light-driven photocatalytic performance, J. Sol-Gel Sci. Technol., 72 (2014) 443-454.

[20] Z.C. Sun, Z.Q. Yu, Y.Y. Liu, C. Shi, M.S. Zhu, A.J. Wang, Construction of 2D/2D BiVO4/g-C3N4 nanosheet heterostructures with improved photocatalytic activity, J. Colloid Interface Sci., 533 (2019) 251-258.

[21] S.J. Dong, G.J. Lee, R. Zhou, J. J. Wu, Synthesis of g-C3N4/BiVO4 heterojunction composites for photocatalytic degradation of nonylphenol ethoxylate, Sep. Purif. Technol., 250 (2020) 8.

[22] N.A. Mohamed, H. Ullah, J. Safaei, A.F. Ismail, M.F.M. Noh, M.F. Soh, M.A. Ibrahim, N.A. Ludin, M.A.M. Teridi, Efficient photoelectrochemical performance of gamma irradiated g-C3N4 and its g-C3N4@BiVO4 heterojunction for Solar water splitting, J. Phys. Chem. C, 123 (2019) 9013-9026.

Figure R2. XRD patterns of {010}BiVO4, {012}BiVO4, {010}BiVO4/0.2g-C3N4 and {012}BiVO4/0.1g-C3N4. (Figure R2 corresponds to Fig. 2a in the revised manucript)

Question 4: All core-level XPS graphs of O1s should supply

Our response: Thanks a lot for your valuable suggestion. The core-level XPS spectra of O 1s were illustrated with two or three peaks of deconvolution and shown Figure R3. Peak at about 530.0 and 532.0 is assigned to the lattice O2- and adsorbed O2- molecules, respectively [23-25]. Interestingly, for {010}BiVO4, a characteristic peak of adsorbed O2- was observed at about 531.4 eV, which can be attribute to the adsorbed -OH [26]. Instead, for {012}BiVO4, the characteristic peak of adsorbed O2- is at about 532.5 eV that originates from the C=O adsorption [27]. These different properties can be ascribed to the diversely exposed crystal planes, which is in accordance with the analysis of FTIR in manuscript. Furthermore, after introducing g-C3N4 substrate, the binding energy of lattice O2- decreased in both {010}BiVO4/0.2g-C3N4 and {012}BiVO4/0.1g-C3N4, indicating the electrons transfer from BiVO4 to g-C3N4 and corresponding to results of XPS spectra of Bi 4f and V 2p. The above result is also supplied in the revised manuscript and supporting information as “red” marked, and the result is as follows:

Futhermore, The high resolution XPS spectra of O 1s were illustrated with two or three peaks of deconvolution in Fig. S1b. Peak at about 530.0 and 532.0 is assigned to the lattice O2- and adsorbed O2- molecules, respectively. Interestingly, for {010}BiVO4, a characteristic peak of adsorbed O2- was observed at about 531.4 eV, which can be attribute to the adsorbed -OH. Instead, for {012}BiVO4, the characteristic peak of adsorbed O2- is at about 532.5 eV that originates from the C=O adsorption. These different properties can ascribe to the diversely exposed crystal planes, which is in accordance with the analysis of FTIR. Furthermore, after introducing g-C3N4 substrate, the binding energy of lattice O2- decreased in both {010}BiVO4/0.2g-C3N4 and {012}BiVO4/0.1g-C3N4, indicating the electrons transfer from BiVO4 to g-C3N4 and corresponding to results of XPS spectra of Bi 4f and V 2p.

Ref:

[23] X. Zhao, Y. Fan, W. Zhang, X. Zhang, D. Han, L. Niu, A. Ivaska, Nanoengineering Construction of Cu2O Nanowire Arrays Encapsulated with g-C3N4 as 3D Spatial Reticulation All-Solid-State Direct Z-Scheme Photocatalysts for Photocatalytic Reduction of Carbon Dioxide, ACS Catalysis, 10(2020) 6367-6376.

[24] X. Feng, X. Zhao, L. Chen, BiVO4/BiO0.67F1.66 heterojunction enhanced charge carrier separation to boost photocatalytic activity. J. Nanopart. Res., 21(2019) 61.

[25] S. S. Mali, G. R. Park, H. Kim, H. H. Kim, J. V. Patil, C. K. Hong, Synthesis of nanoporous Mo:BiVO4 thin film photoanodes using the ultrasonic spray technique for visible-light water splitting, Nanoscale Adv., 1(2019), 799-806.

[26] G. L. Wang, W. Q. Zhang, J. Y. Li, X. L. Dong, X. F. Zhang, Carbon quantum dots decorated BiVO4 quantum tube with enhanced photocatalytic performance for efficient degradation of organic pollutants under visible and near-infrared light. J. Mater. Sci., 54 (2019) 6488–6499.

[27] X. Xu, Q. L. Zou, Y. S. Yuan, F. Y. Ji, Z. H Fan, B. Zhou, Preparation of BiVO4-graphene nanocomposites and their photocatalytic activity, J. Nanomater., 2014(2014) 401697.

Figure R3. High resolution XPS spectra of O1s on {010}BiVO4, {010}BiVO4/0.2g-C3N4, {012}BiVO4 and {012}BiVO4/0.1g-C3N4. ((Figure R3 corresponds to Fig. S1b in the revised supporting information)

Question 5: Photoelectrochemical analysis must supply in the revised MS

Our response: Thanks a lot for your valuable suggestion. As shown in Figure R4, positive tangents in Mott-Schottky plots of {010}BiVO4, {012}BiVO4 and g-C3N4 were observed, which which reveals the n-type semiconductor characteristics of these prepared samples. For n-type semiconductor, the conduction band (ECB) is considered very close to the flat band [30-31]. Hence, by calculating the intercept of tangent line in Mott-Schottky plots, the conduction band energy potentials of {010}BiVO4, {012}BiVO4 and g-C3N4 were estimated as -0.45, -0.24 and -0.07 V vs NHE at pH = 0, respectively. Since the bandgap energy has been confirmed by UV-diffuse absorption spectrum, through formula (R1), the valence band (EVB) potentials of {010}BiVO4, {012}BiVO4 and g-C3N4 were calculated to be 1.96, 2.15 and 2.73 V vs NHE at pH = 0, respectively.

Eg = EVB – ECB              (R1)

These ascensive analysis is also supplied in the revised manuscript as “red” marked, and the result is as follows:

As shown in Fig. 4b, positive tangents in Mott-Schottky plots of {010}BiVO4, {012}BiVO4 and g-C3N4 were observed, which reveals the n-type semiconductor characteristics of these prepared samples. For n-type semiconductor, the conduction band (ECB) is considered very close to the flat band. Hence, by calculating the intercept of tangent line in Mott-Schottky plots, the conduction band energy potentials of {010}BiVO4, {012}BiVO4 and g-C3N4 were estimated as -0.45, -0.24 and -0.07 V vs NHE at pH = 0, respectively. Since the bandgap energy has been confirmed by UV-diffuse absorption spectrum, through formula (2), the valence band (EVB) potentials of {010}BiVO4, {012}BiVO4 and g-C3N4 were calculated to be 1.96, 2.15 and 2.73 V vs NHE at pH = 0, respectively.

Figure R4. Mott-Schottky plots of {010}BiVO4, {012}BiVO4 and g-C3N4. ((Figure R4 corresponds to Fig. 4b in the revised manucript)

Ref:

[28] P. Y. Kuang, Y. Z. Su, G. F. Chen, Z. Luo, S. Y. Xing, N. Li, Z, Q. Liu, g-C3N4 decorated ZnO nanorod arrays for enhanced photoelectrocatalytic performance. Appl. Surf. Sci., 358(2015) 296-303.

[29] J. J. Xu, B. B. Feng, Y. Wang, Y. D. Qi, J. F. Niu, M. D. Chen, BiOCl decorated NaNbO3 nanocubes: a novel p-n heterojunction photocatalyst with Improved activity for ofloxacin degradation, Front. Chem., 6(2015) 393.

[30] M. Zubair, I. H. Svenum, M. Ronning, J. Yang, Core-Shell Nanostructures of graphene-wrapped CdS nanoparticles and TiO2 (CdS@G@TiO2): the role of graphene in enhanced photocatalytic H2 generation, Catalysts, 10(2020) 358.

[31] M. Akple; T. Ishigaki, P. Madhusudan, Bio-inspired honeycomb-like graphitic carbon nitride for enhanced visible light photocatalytic CO2 reduction activity. Environ. Sci. Pollut. R., 27(2017) 1-15.

Reviewer 3 Report

The manuscript from Xiaojing Zhang and co-workers reported the photocatalytic degradation of tetracycline over the (010) or (012) facet bismuth vanadium oxide and carbon nitride. Although, there are already many publications on the photocatalytic performance of the BiVO4 and C3N4 composites, this manuscript provides other points of view, for example the effect of the BiVO4 facets and the clarification of photocatalytic degradation mechanism. However, there are also some editorial or grammatical mistakes that should be revised. Please also verify the letter format. This article can be published after a minor revision.

Author Response

Thanks for your valuable suggestion. We are sorry for the grammar and spelling errors. The manuscript and supporting information both have been carefully proofread with the help of a native English speaker in order to make the article as comprehensible as possible. And all the corrections have been marked in red, which are hoped to meet your approval.

Reviewer 4 Report

In the manuscript " Type Ⅱ heterojunction formed between {010} or {012} facets dominated bismuth vanadium oxide and carbon nitride to enhance the photocatalytic degradation of tetracycline" Ma et al. presents the fabrication of {010} and {012} facets dominated bismuth vanadium oxide, which is decorated by carbon nitride to investigate their heterojunction structure, simultaneously the corresponding photocatalytic efficiencies are tested. The presented results are interesting. This manuscript is well-organized and carefully written. It can be accepted after minor revision. The comments are presented as follows:

1. The latest literature about carbon nitride should be cited, such as Chemical Engineering Journal, 2022, 429, 132388; Chemical Engineering Journal, 2020, 396, 125229; Coordination Chemistry Reviews, 2022, 462, 214500.

2. In “Herein, {010}C and {012} facets dominated bismuth vanadium oxide were rationally fabricated …….,”  C should be deleted. 

3. The font size of the article is inconsistent. Please check it.

4. In Figure 6d, why the degradation rate decreases with the cycle? The {010}BiVO4/g-C3N4 after stability tests should be investigated. 

Author Response

Question 1: The latest literature about carbon nitride should be cited, such as Chemical Engineering Journal, 2022, 429, 132388; Chemical Engineering Journal, 2020, 396, 125229; Coordination Chemistry Reviews, 2022, 462, 214500.

Our response: Thanks a lot for your valuable suggestion, and we have supplied the above references in “Introduction” section as follows:

  1. Wan, C.; Zhou, L.; Xu, S.; Jin, B.; Ge, X.; Qian, X.; Xu, L.; Chen, F.;Zhan, X.; Yang, Y.; Cheng, D. G.;. Defect engineered mesoporous graphitic carbon nitride modified with AgPd nanoparticles for enhanced photocatalytic hydrogen evolution from formic acid, Chem. Eng. J. 2022, 429, 132388.
  2. Wan, C.; Zhou, L.; Sun, L.; Xu, L.; Cheng, D.; Chen, F.; Zhan, X.; Yang, Y. Boosting visible-light-driven hydrogen evolution from for mic acid over AgPd/2D g-C3N4 nanosheets Mott-Schottky photocatalyst, Chem. Eng. J. 2020,396,125229.
  3. Wei, R.; Tang, N.; Jiang, L.; Yang, J.; Guo, J.; Yuan, X.; Liang, J.; Zhu, Y.; Wu, Z.; Li, H. Bimetallic nanoparticles meet polymeric carbon nitride: Fabrications, catalytic applications and perspectives, Coordination Chemistry Reviews, 2022, 462, 214500.

Furthermore, some other latest and significant literatures were also complemented in the “Results and discussion” section, which are summarized in the “References” (Page 15-16 in the revised manuscript).

Question 2: In “Herein, {010}C and {012} facets dominated bismuth vanadium oxide were rationally fabricated …….,”  C should be deleted.

Our response: Thanks a lot for your kind reminding. We are sorry for the spelling errors. We have removed the C letter in the revised manuscript. Meanwhile the manuscript and supporting information both have been carefully proofread. And all the corrections have been marked in red, which are hoped to meet your approval.

Question 3: The font size of the article is inconsistent. Please check it.

Our response: Thanks a lot for your kind reminding. We are sorry for the mistakes. We have revised the font size, and carefully proofread the manuscript and supporting information as well. All the corrections have been marked in red in the revised manuscript, which are expected to attain your approbation.

Question 4: In Figure 6d, why the degradation rate decreases with the cycle? The {010}BiVO4/g-C3N4 after stability tests should be investigated.

Our response: Thanks a lot for your valuable suggestion. As shown in Figure R5a, after 4 cycles, the degradation efficiency of {010}BiVO4/g-C3N4 exhibited merely 3% reduction, which represents a relatively stable performance in the field of photocatalysis [32-34]. The minor decline of degradation efficiency may be ascribed to the attenuate of surface adsorption with recycling. As shown in Figure R5b, the BET spectrum indicate that the specific surface area of {010}BiVO4/g-C3N4 is about 9.6816 m2/g, which represents excellent adsorption performance and can conduce to the photocatalytic activity [35]. However, during the photodegradation of TC, some macromolecular substances should be inevitably absorbed on the surface of {010}BiVO4/g-C3N4 (Figure R5c) and affect the photocatalytic activity, which could not be emancipate by water clean during the recycle experiment. Therefore, a light degradation efficiency reduction emerged with the recycling of photocatalysis [36]. Meanwhile, this is also an evidence for the stability of BiVO4, which is in accordance with the reported investigations [37-38]. Because, if the photocatalyst is unstable, the performance would be reduced significantly as our previous study and other reports [32, 39-40]. This is also an important reason why we focused on BiVO4. The above analysis is also supplied in the revised manuscript and supporting information and marked in “red” marked. The result is as follows:

Furthermore, the cycling experiments showed that after 4 cycles (Fig. 6d), the degradation efficiency of {010}BiVO4/g-C3N4 exhibited merely 3% reduction, which represents a relatively stable performance in the field of photocatalysis. The minor decline of degradation efficiency may be ascribed to the attenuate of surface adsorption with recycling. As shown in Fig. S3, the BET spectrum indicate that the specific surface area of {010}BiVO4/g-C3N4 is about 9.6816 m2/g, which represents excellent adsorption performance and can conduce to the photocatalytic activity. However, during the photodegradation of TC, some macromolecular substances should be inevitably absorbed on the surface of {010}BiVO4/g-C3N4 ((Fig. 7b) and affect the photocatalytic activity, which could not be emancipate by water clean during the recycle experiment. Therefore, a light degradation efficiency reduction emerged with the recycling of photocatalysis.

Figure R5. (a)Stability tests of TC degradation and (b) N2 adsorption-desorption isotherms over {010}BiVO4/g-C3N4, (c) LC-MS spectra of intermediates over {010}BiVO4/0.2g-C3N4 during TC degradation. (Figure R5a, R5b and R5c correspond to Fig. 6d, Fig. S3 and Fig. 7b in the revised manucript and supporting information, respectively)

Ref.

[32] S. Li, B. Xue, C. Wang, W. Jiang, S. Hu, Y. Liu, H. Wang, J. Liu, Facile fabrication of flower-like BiOI/BiOCOOH p–n heterojunctions for highly efficient visible-light-driven photocatalytic removal of harmful antibiotics, Nanomaterials, 9(2019) 1571.

[33] Z. Zhu, F. Liu, H. Zhang, J. Zhang, L. Han, Photocatalytic degradation of 4-chlorophenol over Ag/MFe2O4 (M = Co, Zn, Cu, and Ni) prepared by a modified chemical co-precipitation method: a comparative study, RSC Adv., 5(2015) 55499-55512.

[34] H. Fu, L. Yang, D. Hu, C. Yu, Y. Ling, Y. Xie, S. Li, J. Zhao, Titanium dioxide nano-heterostructure with nanoparticles decorating nanowires for high-performance photocatalysis, Int. J. Hydrogen. Energ., 43(2018) 1-9.

[35] P. Nagaraju, S. H. Puttaiah, K. Wantala, Preparation of modified ZnO nanoparticles for photocatalytic degradation of chlorobenzene, Appl. Water Sci., 10(2020) 137.

[36] M. Ganeshbabu, N. Kannan, P. S. Venkatesh, G. Paulraj, K. Jeganathan, D. MubarakAli, Synthesis and characterization of BiVO4 nanoparticles for environmental applications, RSC Adv., 10 (2020) 18315-18322

[37] U. Lamdab, K. Wetchakun, S. Phanichphant, W. Kangwansupamonkon, N. Wetchakun, Highly efficient visible light-induced photocatalytic degradation of methylene blue over InVO4/BiVO4 composite photocatalyst, J. Mater. Sci., 50(2015) 5788-5798.

[38] M. Guo, Q. He, A. Wang, W. Wang, Z. Fu, A novel, simple and green way to fabricate BiVO4 with excellent photocatalytic activity and its methylene blue decomposition mechanism, Crystals, 6(2016) 81.

[39] X. J. Zhang, X. Zhao, K. Chen, Y. Y. Fan, S. L. Wei, W. S. Zhang, D. X. Han, L. Niu, Palladium-modified cuprous (I) oxide with {100} facets for photocatalytic CO2 reduction, Nanoscale, 13(2021) 2883.

[40] M. Hu, Y. Quan, S. Yang, R. Su, H. Liu, M. Gao, L. Chen, J. Yang, Self-cleaning semiconductor heterojunction substrate: ultrasensitive detection and photocatalytic degradation of organic pollutants for environmental remediation. Microsyst. Nanoeng., 6 (2020) 111.

Round 2

Reviewer 1 Report

Can be publishable.